# Isolation, Identification, and Pathogenicity of a Goose Astrovirus Genotype 1 Strain in Goslings in China

**DOI:** 10.3390/v16040541

**Published:** 2024-03-30

**Authors:** Feng Wei, Dalin He, Bingrong Wu, Youxiang Diao, Yi Tang

**Affiliations:** 1College of Animal Science and Technology, Shandong Agricultural University, 61 Daizong Street, Tai’an 271018, China; 18854888209@163.com (F.W.); dlhe1230@163.com (D.H.); brongwu2021@163.com (B.W.); 2Shandong Provincial Key Laboratory of Animal Biotechnology and Disease Control and Prevention, Tai’an 271018, China; 3Shandong Provincial Engineering Technology Research Center of Animal Disease Control and Prevention, Tai’an 271018, China

**Keywords:** GAstV-1, goslings, pathogenicity, qRT–PCR, viral loads

## Abstract

Goose astrovirus genotype 1 (GAstV-1) has emerged in goose farms in some provinces of China in recent years and is considered to be one of the pathogens of gout in goslings in China. However, few studies have been conducted on the dynamic distribution, tissue tropism, and pathogenesis of GAstV-1 in goslings. In 2022, an epidemiological investigation of goose astrovirus (GAstV) in goslings was conducted in seven provinces of China. During the investigation, a GAstV-1 designated as GAstV-JSXZ was identified in the kidney of an 8-day-old gosling and was successfully isolated from a goose embryo. The full genome sequence of GAstV-JSXZ was determined using the next-generation sequencing technique. The complete genome of GAstV-JSXZ was 7299-nt-long. Interestingly, the phylogenetic analysis revealed that Chinese GAstV-1 has formed two distinct subgroups based on the ORF 2 genomes, designated GAstV-1 1a and GAstV-1 1b. The GAstV-JSXZ shared the highest identity with GAstV-1 1a strain FLX and TZ03 in nucleotides (ORF1a: 98.3–98.4%; ORF1b: 92.3–99.1%; ORF2: 95.8–98.8%) and amino acid sequences (ORF1a: 99.4–99.5%; ORF1b: 98.2–98.8%; ORF2: 97.0–99.4%). To evaluate the pathogenicity of GAstV-1, 1-day-old goslings were inoculated with the virus by oral and subcutaneous injection routes, respectively. The results revealed that the virus causes extensive pathological organ damage, especially in the kidney, liver, and thymus. Virus-specific genomic RNA could be detected in the cloacal swabs and tissues of infected goslings throughout the experiment. The viral copy numbers examined in the kidney and intestine were the highest, followed by the liver and spleen. These results are likely to provide a new understanding of the pathogenicity of GAstV-1 in geese.

## 1. Introduction

Astroviruses (AstVs) are single-stranded, positive-sense, and non-enveloped RNA viruses with a whole genome length of 6.4–7.9 kb that belong to the *Astroviridae* family [1]. Their genome consists of a 5′-untranslated region (UTR), three open reading frames (ORF1a, ORF1b, and ORF2), a 3′-UTR, and a poly-A tail [2]. ORF1a and ORF1b encode two non-structural (NS) polyproteins, whereas ORF2 encodes structural protein involved in receptor recognition and immune response. ORF2 is the most divergent region between serotypes [3,4]. The AstV CP contains four functional domains: an N-terminal domain, a core domain, a spike domain, and an acidic C-terminal domain (ACTD) [5]. AstVs have been isolated from different poultry flocks, including turkeys [6], ducks [7], chickens [8], guinea fowls [9], pigeons [10], and geese [11], and cause various diseases, such as turkey enteritis, duck hepatitis, and goose gout.

Goose astrovirus (GAstV) is divided into two distinct phylogenetic clades (GAstV-1 and GAstV-2) based on the ORF2 region [12]. Since 2017, several reports have confirmed that GAstV-2 is one of the pathogens responsible for gosling gout diseases, which have brought considerable economic losses to the Chinese goose industry [11,13,14,15] since GAstV-1 (GAstV FLX) was first reported in 2017 in Hunan Province, China. The full lengths of GAstV FLX were 7299 nucleotides (nt) and the genomic organisation was similar to those of other known astroviruses [16]. Several GAstV-1 strains, such as TZ03, AHDY, SCCD, ZJC14, JXYC, and JXGZ, have been sequenced (Appendix A). The viral pathogenicity of GAstV-1 (TZ03 strain) was first determined in 2021 and is considered to be one of the causative agents of gosling gout in China.

In this study, an epidemiological investigation on goslings was conducted in seven provinces of China and an astrovirus designated as GAstV-JSXZ was successfully isolated from the goose embryo. In addition, specific primers and probes were designed according to ORF2 of GAstV-1 and the reaction conditions were optimised. A TaqMan-based one-step real-time quantitative reverse transcriptase–polymerase chain reaction (qRT–PCR) method was established for detecting GAstV-1. The complete viral genomes of the newly identified GAstV-1 were sequenced and analysed and experimental infection studies were performed to validate the pathogenicity of the isolated strain.

## 2. Materials and Methods

### 2.1. Ethics Statement

All animal infection experiments were conducted in accordance with the “Guidelines for Experimental Animals” of the Ministry of Science and Technology (Beijing, China) and were conducted under the supervision of the Animal Protection and Utilization Committee of the Shandong Agricultural University.

### 2.2. Sample Collection

A total of 187 samples (e.g., livers, kidneys, and dead goose embryos) were collected from 10 regions in seven Chinese provinces during 2022 (Table 1). Each tissue taken from the same animal was counted as one sample. The tissue samples (e.g., livers and kidneys) and goose embryos (e.g., allantoic fluids, embryo bodies, and yolk membranes) were pooled and homogenised for further analysis. Viral DNA/RNA was extracted using a TransGen virus DNA/RNA kit (TransGen Biotech, Beijing, China) following the manufacturer’s directions for all samples.

### 2.3. Virus Isolation

The goose embryos were purchased from a healthy goose farm in Shandong Province, China, and tests for the waterfowl-origin viruses (including TMUV, GPV, GHPV, AIV, GRV, and GAstV) were all negative (Appendix A). For virus isolation, homogenated kidney samples (Xuzhou County) that tested positive for GAstV-1 were selected and inoculated into the chorioallantoic cavity of 11-day-old goose embryos. The goose embryos were monitored daily. After several serial passages, the allantoic fluids were harvested within 3–5 days post inoculation for the detection of GAstV-1 and, then, preserved at −80 °C as viral stocks.

### 2.4. Next-Generation Sequencing

The total RNA was extracted from a GAstV-1 strain which was named GAstV-JSXZ (Xuzhou County) using a TruSeq stranded total RNA sample prep kit (Illumina, San Diego, CA, USA) to build the RNA library. The full-length genome of the GAstV-JSXZ was investigated by next-generation sequencing (NGS) with the Hiseq platform (Illumina) in accordance with the manufacturer’s protocol (Shanghai Personal Biotechnology Co., Ltd., Shanghai, China). The analysis and de novo assembly of all NGS raw data were the same as described by Wei et al. [17].

### 2.5. Phylogenetic Analysis

To determine the genetic relationships of the GAstV-JSXZ with the known sequences of other GAstV and AstVs, the identities of the nucleotide and deduced amino acid (aa) sequences of the isolated strain were aligned using the ClustalW matrix (MegAlign) with Lasergene 7.0 software. Phylogenetic trees of the three ORFs were constructed via the maximum likelihood (ML) method implemented in the MEGA version 6.0 program by taking 1000 bootstrap replications [18].

### 2.6. Experimental Infection Study

To investigate the pathogenicity of the isolated GAstV-JSXZ, 120 healthy 1-day-old goslings (Tai’an, China), which were free of GAstV-1-specific nucleic and other waterfowl-origin viruses, were divided into three groups (A, B, and C), each including 40 birds and reared in different negative pressured isolators. A total of 80 goslings were inoculated with 0.2 mL viral fluid (GAstV-JSXZ strain; 10^4.25^ ELD_50_/0.2 mL) via oral infection (Group A) or subcutaneous injection (Group B), while the other goslings were inoculated with 0.2 mL sterile PBS in the same manner as the uninfected control (Group C). Clinical signs and mortality were monitored and recorded daily.

### 2.7. Establishment of qRT–PCR for Detection of GAstV-1

The viral load of the collected samples was determined by the qRT–PCR method established in this study. The pair of specific primers and TaqMan probes were designed to target the ORF2 gene of GAstV-1 (Table 2). RNA was extracted from the GAstV-1 allantoic fluid and then reverse-transcribed into cDNA. GAstV-1-F/R was used for PCR amplification. The PCR product was separated, purified, and ligated to the pMD18-T vector (Promega, Madison, WI, USA) to construct standard plasmids (GAstV-1-ORF2). Then, qRT–PCR was performed using the GoldStar probe one-step RT-qPCR kit (CWBIO, Taizhou, China) in a 25 μL reaction system. The optimised reaction program was as follows: 2 μL total RNA, 1 μL GoldStar probe one-step EnzymeMix, 12.5 μL 2 × GoldStar probe one-step buffer, 1 μL RNA template, 0.5 mL (0.2 μmol/L) forward and reverse primers and probes, and the remainder was made up to 25 μL with RNase-free ddH_2_O. Then, qRT–PCR was performed using the Roche LightCycler 96 real-time PCR system (Roche, Basel City, Switzerland) under the following optimised reaction conditions: 45 °C for 10 min, reverse transcriptase inactivation at 95 °C for 10 s, followed by 45 cycles at 95 °C for 5 s and at 60 °C for 20 s and ending at 4 °C. The standard curve and cycle threshold (CT) values were analysed by sequence detector software (version 2.1, Applied Biosystems, Waltham, MA, USA). The standard plasmid was serially diluted 10-fold and 1.0 × 10^8^~1.0 × 10^1^ copies/µL of standard plasmid was used as a template for qRT–PCR amplification under optimised reaction conditions. The results were analysed and used to construct a standard curve. To determine the sensitivity of the assay, standard plasmid was serially diluted 10-fold at a concentration of 1.0 × 10^8^~1.0 × 10^1^ copies/µL and amplified under optimised reaction conditions. At the same time, conventional RT–PCR was performed to compare the sensitivity of qRT–PCR. The qRT–PCR assay was performed with different waterfowl viruses (GPV, AIV, NDV, ARV, GAstV-2, and FAV) to evaluate the specificity of the reaction. The repeatability test was performed with the standard plasmid for GAstV-1-ORF2 at the concentration of 1.0 × 10^8^~1.0 × 10^1^ copies/µL via optimised protocol. The accuracy of the method was evaluated by calculating the intra-assay and inter-assay standard deviations. Each dilution gradient was repeated in 3 wells and ddH_2_O was used as a negative control.

### 2.8. Sample Collection

At 2, 4, 6, 8, 10, 12, 14, and 16 days post inoculation (dpi), three goslings (which did not die after infection) in each group were randomly selected for weighing. At the same time, cloacal swabs were collected from all goslings and, then, the goslings were euthanised with intravenous pentobarbital sodium and necropsied. The cloacal swabs and tissue samples (including the hearts, livers, spleens, lungs, kidneys, bursas, thymuses, pancreas, brains, proventriculuses, and intestines) of all goslings were collected and viral load was determined by qRT–PCR. The tissues of the goslings were examined for histopathological observation and identification.

## 3. Results

### 3.1. Viral Nucleic Acids Detection in the Clinical Samples

In 2022, 187 samples were collected by our team in seven Chinese provinces. The kidney samples collected from Jiangsu Province (Xuzhou County) were identified as GAstV-1-positive by RT–PCR and free of other waterfowl-origin viruses. These kidney samples were used for virus isolation. A total of 42.8% (15/35) of the GAstV-1-positive samples were co-infected with GAstV-2. A total of 51.4% (18/35) of GAstV-2-positive samples were co-infected with AIV; 35.0% (7/20) were co-infected with GRV. In addition, 119 of 187 samples (63.6%) tested positive for GAstV-2 by RT–PCR. Details of viral nucleic acid detection in clinical samples are shown in Table 1.

### 3.2. Virus Isolation

To isolate the GAstV-1 strain, the homogenated kidney tissues from Jiangsu Province (Xuzhou County) were prepared to be inoculated into goose embryos. After three passages, one GAstV-JSXZ strain (GenBank accession no. OR827024) was successfully isolated from the samples and confirmed to be positive for GAstV-1 and negative for other waterfowl-origin viruses. In total, 40% of inoculated goose embryos died after incubation within 72–96 h and severe diffuse haemorrhages of the dead goose embryos could be observed (Figure 1).

### 3.3. NGS Analysis

A total of 2,255,752 sequencing reads of 151 mer were generated on the HiSeq sequencer in 1.03 Gb of FastQ format sequence data from extracted viral stocks. After being processed by quality control (QC) filters of the Hiseq platform, low-quality reads, trim poly-T tails, and adapter sequences were removed. As a result, the residual 231,327 (10.2%) reads were considered clean reads and assembled de novo using CLC genomic workbench software (version 10.0). Finally, a single long contig from the remapping reads that covered the full length of astrovirus reference sequences was obtained and no other waterfowl-origin viral sequences were discovered by analysing all assembled contigs.

### 3.4. Phylogenetic Analysis

To further investigate the evolutionary relationships between GAstV-JSXZ and other representative avian members, a phylogenetic tree was constructed based on the deduced nucleotide sequences of the ORF1a, ORF1b, and ORF2 homologous genes with MEGA 6.0 software. As shown in Figure 2, all three trees demonstrated that the GAstV-JSXZ was clustered together with the GAstV-1 strains, including FLX, TZ03, ZJC14, AHDY, JXYC, and SCCD strains. The deduced nucleotide sequences from the ORF1a and ORF2 phylogenetic tree indicate that the GAstV-1 was adjacent to the DAstV-1 strains, while GAstV-1 formed another clade with TAstV-1 in the ORF1b region. In addition, the phylogenetic tree based on the ORF2 genomes of eight GAstV-1 strains revealed two subgroups designated GAstV-1 1a and GAstV-1 1b, based on genotype. The deduced amino acid sequences of the three ORFs and the full-length genome sequence of the GAstV-JSXZ were compared with other representative avian members. As described in Table 3, the GAstV-JSXZ was highly similar to the GAstV-1 strain in terms of the nucleotide (ORF1a: 98.1–98.8%, ORF1b: 92.3–99.1%) and aa sequence (ORF1a: 99.4–99.7%, ORF1b: 98.2–99.8%). The ORF2 sequence was shown to be a variable region, in which the GAstV-JSXZ shared the highest identity with the GAstV-1 1a strain FLX and TZ03 in terms of the nucleotide (95.8–98.8%) and aa sequences (97.0–99.4%), and displayed low identity with the GAstV-1 1b strain with respect to the nucleotide (74.6–75.0%) and aa sequences (80.7–81.0%), including ZJC14, AHDY, JXYC, and SCCD strains. In a further analysis, the GAstV-JSXZ had low similarity to other representative avian members in the nucleotide (ORF1a: 50.4–58.8%; ORF1b: 59.1–65.8%; ORF2: 45.5–55.2%) and aa sequences (ORF1a: 41.2–55.3%; ORF1b: 59.6–66.5%; ORF2: 36.5–52.5%).

### 3.5. Establishment of qRT–PCR for Detection of GAstV-1

To determine the viral load of the collected samples, qRT–PCR was performed in this study. The results showed that the standard curve showed a good linear relationship in the range of 1.0 × 10^8^~1.0 × 10^1^ copies/µL plasmid template, with a slope of −3.926. The standard formula is y = −3.926x + 41.268 and (R^2^) of 0.9971. The minimum detection limit for plasmid standards is 10 copies/µL and the sensitivity of conventional RT–PCR was 10^4^ copies/µL (Figure 3). The specificity test showed that the method did not cross-react with GPV, AIV, NDV, ARV, GAstV2, or FAV. The intra-assay and inter-assay coefficients of variation were 10 copies/µL and 1.0 × 10^4^ copies/µL, respectively. The intra-assay and inter-assay standard deviations (SD) ranged from 0.11 to 0.21 and from 0.07 to 0.22, respectively. The intra-assay and inter-assay coefficients of variation (CV) were both within 1.0%. Therefore, the qRT–PCR method established in this study can provide reliable technical support for the detection of GAstV-1.

### 3.6. Pathogenicity of GAstV-JSXZ in Goslings

In groups A and B, depression and loss of appetite were observed between 2 and 6 dpi. Twelve goslings in group A died between 2 and 10 dpi, while goslings in group B started to die at 4 dpi with a mortality rate of 32.5% (Figure 3). The infected goslings grew more slowly and gained less weight than the goslings in the control group (Figure 4). The autopsy results showed significant pathological changes in the kidneys of the infected group, including mesenteric haemorrhages and swellings (Figure 5A). There were also signs of haemorrhage in the thymuses and livers of the dead goslings (Figure 5B,C). No mortality or clinical symptoms were observed in the control group (Figure 5a,c). Histopathological analysis revealed significant microscopic changes in the infected goslings. The most obvious microscopic histopathological changes were in the livers, kidneys, and thymuses. The livers showed necrosis and haemorrhage (Figure 6A). The kidneys exhibited renal interstitial haemorrhage, renal tubular epithelial necrosis, and shedding (Figure 6B). Lesions in the thymus were characterised by erythrocyte exudation in the medulla and cortex (Figure 6C). No lesions were observed in the corresponding tissues of the goslings in the control group.

### 3.7. Viral Loads in Cloacal Swabs and Tissues

Viral copies could be detected in the cloacal swabs of the infection groups as early as 2 dpi. The viral load in the infected groups reached a peak value at 8 dpi. Moreover, the subcutaneous injection group had a higher viral load than the oral infection group (Figure 7). The viral RNA could be detected in all of the samples throughout the experiment and attained high levels in all 11 tissues at 6 and 10 dpi (Figure 8). No positive viral RNA was detected in the control group.

## 4. Discussion

In the past two years, the clinical cases of GAstV-1 infection in China have gradually increased, which has attracted people’s attention and reports. Although some studies have been conducted on the isolation and characterization of GAstV-1 viruses, there have been no detailed studies on the pathogenicity of GAstV-1 in goslings [19]. In the present study, a systematic trial of the pathogenicity of GAstV-1 in goslings has been carried out.

Due to the absence of biosafety protection measures and the change in breeding mode, the outbreak of infectious diseases in China’s goose industry has been on the rise in recent years. For a better understanding of the prevalent trends of contagious pathogens (including TMUV, GPV, GHPV, AIV, GRV, and GAstV) in the field, an epidemiological investigation on goslings was conducted in this study. The results showed that the positive rate of GAstV-2 in the samples was as high as 63.6%, indicating that GAstV-2 is still an important factor endangering the healthy development of China’s goose industry. Vertical transmission and the lack of a targeted vaccine are a major reason for the high prevalence of GAstV-2. Several astroviruses have been found to transmit vertically. Chicken astroviruses (CAstV) can be transmitted by vertical transmission and the presence of CAstV can be detected in both just-hatched chicks and dead-shell embryos [20,21]. The GAstV-2 is capable of vertical transmission from breeding geese to goslings and leads to reduced fertilization and hatchability of embryos [22]. Liu et al. confirmed the horizontal and vertical transmission of duck astrovirus CPH by testing fresh duck manure and embryos from hatcheries in different provinces of China [23]. It was noteworthy that GAstV-1 infections were detected in both goslings and dead goose embryos, demonstrating potential vertical transmission of GAstV-2. However, the exact mechanism of vertical transmission of astrovirus is not clear and needs further research to confirm.

GAstV-1 was first discovered in domestic geese in Hunan Province in 2017 and its potential pathogenesis and molecular epidemiology have not been investigated. The process of qRT–PCR is increasingly used for virus detection and quantification due to its rapidity, simplicity, and repeatability. In this study, specific primers and TaqMan probes for the GAstV-1 conserved region were designed and a qRT–PCR detection method was established. The results showed that qRT–PCR had a good linear relationship with the standard curve. The specificity test showed that amplification of GPV, AIV, NDV, ARV, GAstV2, and FAV had no results. The sensitivity was 1000-fold higher than conventional RT–PCR. The intra-assay and inter-assay coefficients of variation were both within 1% of conventional RT–PCR. Therefore, the qRT–PCR method established in this study can specifically, rapidly, and quantitatively measure GAstV-1. Compared with traditional detection methods, the proposed method requires a smaller sample size and higher detection sensitivity, and can detect the presence of the virus at the early stage of virus infection or when the virus concentration is low, which can provide a detection basis for the diagnosis and prevention of GAstV-1.

In this study, the peak of mortality in infected goslings was 4–10 dpi, with a mortality rate as high as 32.5%, which is consistent with previous studies reported by Wang in 2021 and suggests that GAstV-1 has a strong pathogenic effect on goslings [19]. The excretion of uric acid is mainly controlled by a group of transport molecules located at the apex and basolateral side of the renal proximal tubules, including the organic anion transporter (OAT1), the ATP-binding cassette super family (ABC), and the multidrug resistance-associated protein 4 (MRP4) [24,25]. In experimental groups, GAstV-1 may cause renal ureteral obstruction by damaging proximal renal tubular cells and leading to uric acid metabolism. Related to the qRT–PCR data, viral RNA was detected in all tissues examined in the infection groups throughout the experiment, suggesting that GAstV-1 has extensive tissue tropism. Furthermore, kidney tissue had the highest concentration of GAstV-1, suggesting that it may be the primary target organ, which is consistent with previous studies on GAstV-2 infection [26]. GAstV-1 was first detected in goslings with enteritis in Hunan Province, China, but was unsuccessfully isolated due to the lack of efficient in vitro culture techniques [16]. Notably, a high viral copy number was found in the gut of the GAstV-1 infected group, suggesting that the gut is also an important organ for viral invasion and replication. However, no significant lesions were found in the gut in this study, possibly because the histological changes and inflammatory response of the host’s gut to astrovirus infection were relatively low [27]. In addition, the kidney and intestine can be used as options for GAstV-1 detection due to the high viral copy numbers in these tissues.

Taken together, GAstV-2 is still one of the most important pathogens threatening the goose industry in China. At the same time, the pathogenicity of GAstV-1 infection in goslings cannot be ignored, especially the potential vertical transmission of the virus, which makes the prevention and control of the disease more difficult. Therefore, continued surveillance of the prevalence of GAstV-1 in Chinese goose flocks is required.

## Figures and Tables

**Figure 1 viruses-16-00541-f001:**
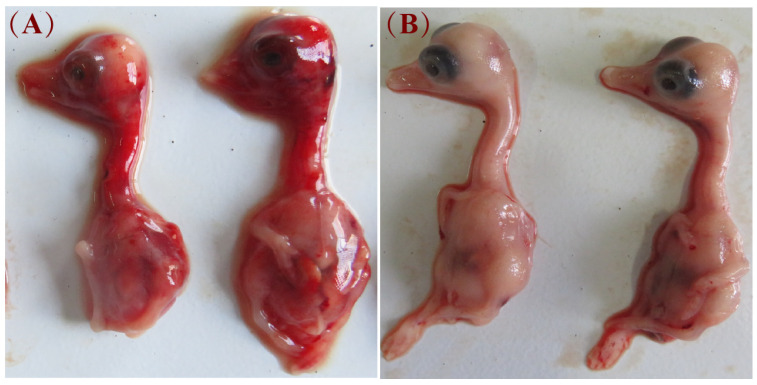
Pathological changes in goose embryos infected with GAstV-1. (**A**) Infected goose embryos at 72 h post infection showing hemorrhage; (**B**) uninfected control goose embryos.

**Figure 2 viruses-16-00541-f002:**
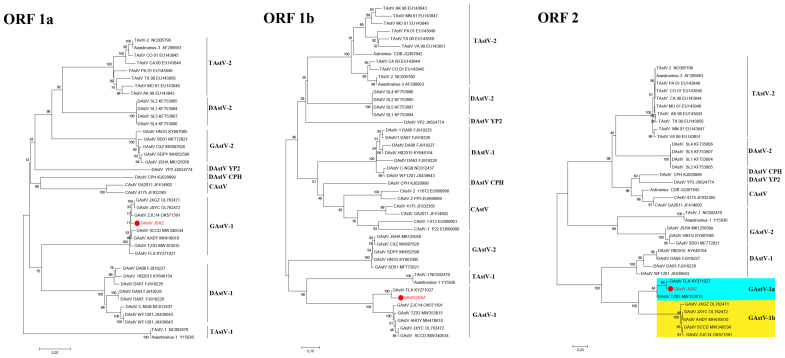
Phylogenetic relationship analysis based on nucleotide sequences of ORF1a, ORF1b, and ORF2 of the GAstV-JSXZ (●) and other AstVs. The trees were generated using MEGA 6.0 software and the neighbour-joining method with 1000 bootstrap replicates. The GAstV-JSXZ isolate determined in this work is indicated by a red dot.

**Figure 3 viruses-16-00541-f003:**
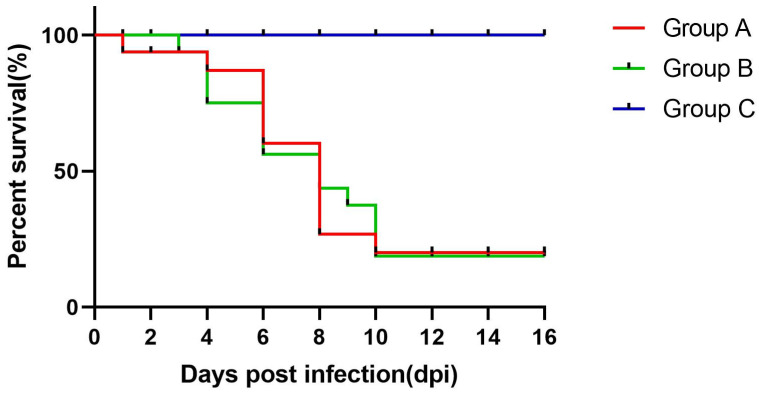
Survival rates of goslings after inoculation with GAstV-JSXZ.

**Figure 4 viruses-16-00541-f004:**
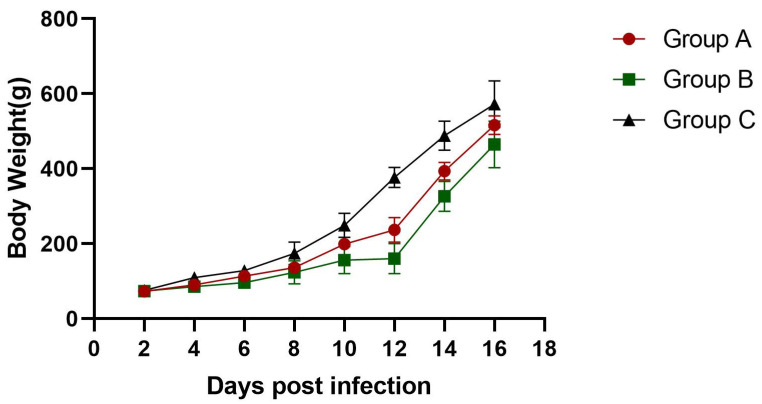
Weight changes in goslings after infection with GAstV-JSXZ.

**Figure 5 viruses-16-00541-f005:**
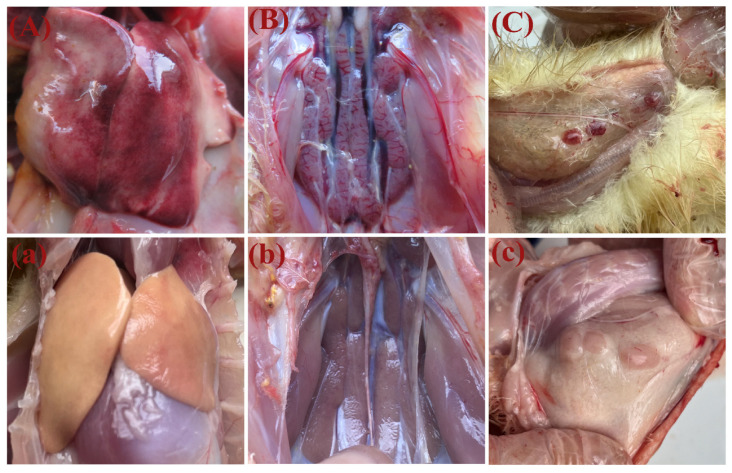
Post-mortem lesions of goslings that died at 6 dpi. (**A**) Hemorrhage and enlargement of the liver; (**B**) severe hemorrhage and swelling of the kidneys; (**C**) severe hemorrhage of the thymus; (**a**–**c**) no clinical symptoms were found in the control group.

**Figure 6 viruses-16-00541-f006:**
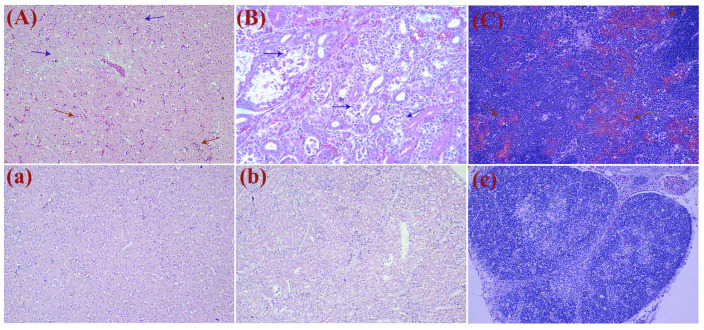
Histopathology lesions of tissues from dead goslings infected with GAstV-1 at 6 dpi. (**A**) H&E-stained liver section showing hepatocyte necrosis (blue arrow) and interstitial bleeding (red arrow); (**B**) H&E-stained kidney section showing the collapse of renal tubular epithelial cells (blue arrow); (**C**) exudate erythrocyte in the medulla and cortex of the thymus (red arrows); (**a**–**c**) normal tissue control. Tissues were stained with hematoxylin and eosin (H&E, 200×).

**Figure 7 viruses-16-00541-f007:**
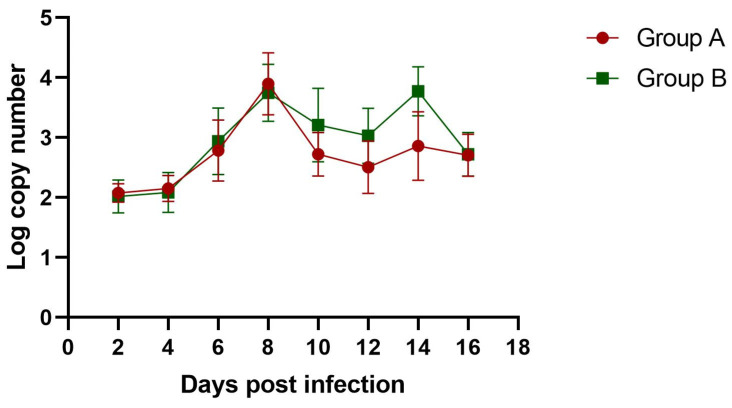
The viral copy numbers in cloacal swabs of goslings infected with GAstV-1.

**Figure 8 viruses-16-00541-f008:**
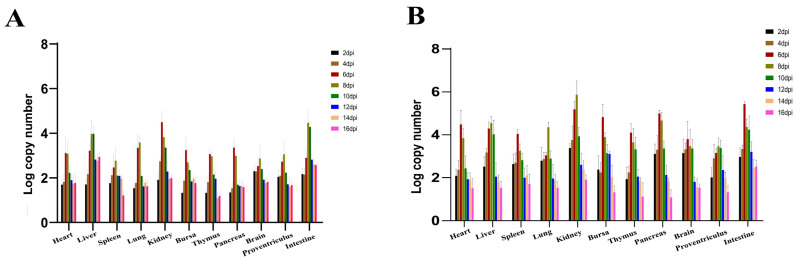
Dynamic distribution of GAstV-1 in experimentally infected goslings. (**A**) The viral copies in tissue samples of the oral infection group; (**B**) the viral copy numbers in tissue samples of the subcutaneous infection group.

**Table 1 viruses-16-00541-t001:** Results of viral nucleic acid detection in clinical samples.

Date	Location	Host	Tissue	Nucleic Acid Detection
March 2022	Sichuan (Chengdu)	Langdes goose	liver	H9 (7/20), GAstV-2 (17/20)
March 2022	Shandong (Jining)	Wulong goose	liver, kidney	GAstV-2 (13/20)
May 2022	Shandong (Liaocheng)	Shitou goose	liver	H9 (4/15), GRV (6/15)
March 2022	Shandong (Heze)	Wulong goose	embryo	GAstV-1 (17/35), GAstV-2 (33/35)
September 2022	Jiangsu (Xuzhou)	Langdes goose	liver	GAstV-1 (10/12)
November 2022	Jiangsu (Suqian)	Zhedong goose	liver, kidney	GAstV-2 (9/15)
April 2022	Henan (Xiangcheng)	Taihu goose	liver, kidney	H9 (9/20), GRV (11/20)
November 2022	Jiangxi (Zhuzhou)	Shitou goose	liver, kidney	GRV (9/20), GAstV-2 (16/20)
December 2022	Jilin (Changchun)	Wulong goose	liver, kidney	GAstV-2 (17/20)
November 2022	Hebei (Xingtai)	Langdes goose	liver, kidney	H9 (13/15), GAstV-2 (14/15)

**Table 2 viruses-16-00541-t002:** Primers used in this study.

Name	Sequence of Primers (5′ to 3′)	Length of the Amplification Products (bp)
GAstV -F	TGAACAGCGTTGATGGAGAT	110 bp
GAstV -Probe	FAM-TCTTCTTCGGACAGCCAATCGCAACCA-BHQ	
GAstV -R	TCACATTTGTTCCCATAGC	

**Table 3 viruses-16-00541-t003:** Sequence identities between GAstV-JSXZ with selected representative astroviruses.

			Sequence Identity (%)					
			ORF1a		ORF1b		ORF2	
Species	GenBank Accession Numbers	Virus Strain	nt	aa	nt	aa	nt	aa
GAstV-1	NC_034567	FLX	98.3	99.5	99.1	98.2	98.8	99.4
	MW353015	TZ03	98.4	99.4	92.3	98.8	95.8	97.0
	OL762471	JXGZ	98.2	99.7	92.8	98.4	75.0	81.0
	OL762472	JXYC	98.1	99.7	92.8	98.6	74.9	80.7
	MH410610	AHDY	98.8	99.5	99.3	99.8	74.6	80.7
	MW340534	SCCD	98.3	99.4	93.0	98.4	74.7	80.8
	OK571391	ZJC14	98.6	99.6	93.5	98.8	74.8	80.8
GAstV-2	MK125058	JSHA	54.1	47.8	64.6	60.8	54.9	42.5
	OL762473	JXGZ	57.1	47.5	64.1	60.8	54.7	42.2
DAstV	FJ919227	DA08	58.8	55.3	64.2	63.7	46.8	37.9
	KF753805	SL2	54.9	48.5	65.8	66.1	51.7	52.5
	JX624774	YP2	52.1	43.5	61.9	59.6	46.8	38.4
	KJ020899	CPH	54.6	48.7	64.3	63.4	45.5	36.5
TAstV	NC002470	TA1	50.4	41.2	59.1	57.2	55.2	48.7
	EU143849	PA01	55.8	48.2	63.8	63.7	47.3	39.0
CAstV	JF414802	GA2011	55.3	50.7	64.4	66.5	48.8	39.2

## Data Availability

The data presented in this study are available on request from the corresponding author upon reasonable request.

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
