# Peer review of "Isolation, Identification, and Pathogenicity of a Goose Astrovirus Genotype 1 Strain in Goslings in China"

_viruses, 2024, doi:10.3390/v16040541_

Round 1
Reviewer 1 Report
Comments and Suggestions for Authors
1、Please label the locations of the lesions in Figure 6 with arrows, and also use different coloured arrows to label different lesions (e.g. necrosis or haemorrhage).
2、Figure 2 is not clear, the strain name is not readable.
3、Table 3 Supplementary strain Genbank numbers. Please add the strains GoAstV-I JXGZ (OL762471) and GoAstV-II JXGZ (OL762473) in table 3, while I could not find the JXGC strain through my review, please double check for any writing errors.
4、The discussion of GoAstV-I pathogenicity studies is not comprehensive enough; please provide a detailed discussion. For example: 10.1016/j.psj.2022.101800.
5、The ORF1b amino acid sequences of the JXGC, JXYC, and TA strains in Table 3 are deducible to calculate amino acid homology, so please make additions.
Comments on the Quality of English Languagegood
Author Response
Dear Editors and Reviewers:
Thank you for your letter and for the reviewers’ comments concerning our manuscript entitled “Pathogenicity of a goose astrovirus genotype 1 strain in goslings, China”. Those comments are all valuable and very helpful for revising and improving our paper, as well as the important guiding significance to our researches. We have studied comments carefully and have made correction which we hope meet with approval. Revised portion are highlighted in yellow in the paper. The main corrections in the paper and the responds to the reviewer’s comments are as flowing:
Reviewer: 1
1、 Please label the locations of the lesions in Figure 6 with arrows, and also use different coloured arrows to label different lesions (e.g. necrosis or hcemorrhage).
Answer: Revised. we had marked it.
2、Figure 2 is not clear, the strain name is not readable.
Answer: Revised. We have submitted high-quality original Figures to the editorial office.
3、Table 3 Supplementary strain Genbank numbers. Please add the strains GoAstV-I JXGZ (OL762471) and GoAstV-II JXGZ (OL762473) in table 3, while I could not find the JXGC strain through my review, please double check for any writing errors.
Answer: Revised.
4、The discussion of GoAstV-I pathogenicity studies is not comprehensive enough; please provide a detailed discussion. For example: 10.1016/j.psj.2022.101800.
Answer: Revised.
5、The ORF1b amino acid sequences of the JXGC, JXYC, and TA strains in Table 3 are deducible to calculate amino acid homology, so please make additions.
Answer: Revised. we had marked it.
Special thanks to you for your good comments.
We tried our best to improve the manuscript and made some changes in the manuscript. These changes will not influence the content and framework of the paper. And here we did not list the changes but marked in yellow in revised paper.
We appreciate for Editors/Reviewers’ warm work earnestly, and hope that the correction will meet with approval.
Once again, thank you very much for your comments and suggestions.
Corresponding author:
Yi Tang
tyck288@sdau.edu.cn
Reviewer 2 Report
Comments and Suggestions for Authors
The article “Pathogenicity of a Goose Astrovirus Genotype I Strain in Goslings, China by Wei et al. identifies the complete genomes and in-vivo pathogenicity of the kidney-associated GAstV-I. comments are as follows:
- line 27: “the virus causes” not “infected goslings cause”.
- line 70: the word “artificial” is not appropriate.
- line 149-150: same information mentioned in line 144-145, why repeated.
- lines 153-157: it seems like SOP not material and methods, please rewrite appropriately.
- lines 160-168: why were the tissues of dead gooselings were not subjected to viral load detection, similarly, why tissues of euthanized goslings were not subjected to histopathology, please explain.
- lines 179-185: what about the virus detection in goslings’ embryos, did the authors try to detect it?
- lines 237-246: do the authors mean this is the result standardized method for qRT-PCR that their study recommends, if so, please clearly indicate, and indicate the specificity and sensitivity %
- MAJOR CONCERN: lines 248-263: it seems that authors only reported lesions in dead goslings, but nothing in the euthanized goslings. this may raise concern regarding the cause of death and its relation to astro viruses, considering the high virus load detected in the euthanized one.
- Figure 6. I would recommend describing the lesions briefly and indicating that these were taken from dead goslings.
- The discussion section is poorly written and does not explain many findings. explanation of high mortality observed, why the high copy number in kidneys...etc
- line 333, is it intramuscular or subcutaneous, please clarify.
Comments on the Quality of English Languageediting of English language required
Author Response
Dear Editors and Reviewers:
Thank you for your letter and for the reviewers’ comments concerning our manuscript entitled “Pathogenicity of a goose astrovirus genotype 1 strain in goslings, China”. Those comments are all valuable and very helpful for revising and improving our paper, as well as the important guiding significance to our researches. We have studied comments carefully and have made correction which we hope meet with approval. Revised portion are highlighted in yellow in the paper. The main corrections in the paper and the responds to the reviewer’s comments are as flowing:
The article “Pathogenicity of a Goose Astrovirus Genotype I Strain in Goslings, China by Wei et al. identifies the complete genomes and in-vivo pathogenicity of the kidney-associated GAstV-I. comments are as follows:
line 27: “the virus causes” not “infected goslings cause”.
Answer: Revised. we had marked it.
line 70: the word “artificial” is not appropriate.
Answer: Revised. we had marked it.
line 149-150: same information mentioned in line 144-145, why repeated.
Answer: Revised. The misrepresentation has been corrected.
lines 153-157: it seems like SOP not material and methods, please rewrite appropriately.
Answer: Revised. we had marked it.
lines 160-168: why were the tissues of dead gooselings were not subjected to viral load detection, similarly, why tissues of euthanized goslings were not subjected to histopathology, please explain.
Answer: Revised. The misrepresentation has been corrected. In this study, tissue samples were collected from all gosling and viral loads were measured. For the sake of data continuity, only data at a specific point in time were presented in this study. Tissue sections were prepared from all infected/uninfected goslings, and some of the observation results were selected for display and annotation in this study.
lines 179-185: what about the virus detection in goslings’ embryos, did the authors try to detect it?
Answer: We collected liver and kidney tissue of the dead goose embryo, and PCR test results showed that GAstV-1 could be detected in both liver and kidney tissues.
lines 237-246: do the authors mean this is the result standardized method for qRT-PCR that their study recommends, if so, please clearly indicate, and indicate the specificity and sensitivity %
Answer: Revised. Please see the attachment for specific results
MAJOR CONCERN: lines 248-263: it seems that authors only reported lesions in dead goslings, but nothing in the euthanized goslings. this may raise concern regarding the cause of death and its relation to astro viruses, considering the high virus load detected in the euthanized one.
Answer: Because the clinical symptoms of euthanized goslings were basically the same as those of dead goslings, we presented only partial results at random. The results of partial euthanasia of goslings are shown below.
-Figure 6. I would recommend describing the lesions briefly and indicating that these were taken from dead goslings.
Answer: Revised. we had marked it.
The discussion section is poorly written and does not explain many findings. explanation of high mortality observed, why the high copy number in kidneys...etc
Answer: Revised. we had marked it.
line 333, is it intramuscular or subcutaneous, please clarify.
Answer: Revised. The misrepresentation has been corrected.
Special thanks to you for your good comments.
We tried our best to improve the manuscript and made some changes in the manuscript. These changes will not influence the content and framework of the paper. And here we did not list the changes but marked in yellow in revised paper.
We appreciate for Editors/Reviewers’ warm work earnestly, and hope that the correction will meet with approval.
Once again, thank you very much for your comments and suggestions.
Corresponding author:
Yi Tang
tyck288@sdau.edu.cn

Round 2
Reviewer 2 Report
Comments and Suggestions for Authors
Accepted